# Propulsion Mechanisms in Magnetic Microrobotics: From Single Microrobots to Swarms

**DOI:** 10.3390/mi16020181

**Published:** 2025-01-31

**Authors:** Lanlan Jia, Guangfei Su, Mengyu Zhang, Qi Wen, Lihong Wang, Junyang Li

**Affiliations:** School of Electronic Engineering, Ocean University of China, Qingdao 266000, China; jialanlan@stu.ouc.edu.cn (L.J.); suguangfei@stu.ouc.edu.cn (G.S.); zhangmengyu6119@stu.ouc.edu.cn (M.Z.)

**Keywords:** microrobot, swarm, magnet, propulsion mechanism, application

## Abstract

Microrobots with different structures can exhibit multiple propulsion mechanisms under external magnetic fields. Swarms dynamically assembled by microrobots inherit the advantages of single microrobots, such as degradability and small dimensions, while also offering benefits like scalability and high flexibility. With control of magnetic fields, these swarms demonstrate diverse propulsion mechanisms and can perform precise actions in complex environments. Therefore, the relationship between single microrobots and their swarms is a significant area of study. This paper reviews the relationship between single microrobots and swarms by examining the structural design, control methods, propulsion mechanisms, and practical applications. At first, we introduce the structural design of microrobots, including materials and manufacturing methods. Then, we describe magnetic field generation systems, including gradient, rotating, and oscillating magnetic fields, and their characteristics. Next, we analyze the propulsion mechanisms of individual microrobots and the way microrobots dynamically assemble into a swarm under an external magnetic field, which illustrates the relationship between single microrobots and swarms. Finally, we discuss the application of different swarm propulsion mechanisms in water purification and targeted delivery, summarize current challenges and future work, and explore future directions.

## 1. Introduction

In recent years, the field of microrobotics has seen many advancements in theory and technology, demonstrating significant flexibility, adaptability, and safety at the micro/nanoscale due to the robots’ small dimensions. For example, in the medical field, microrobots can access complex, variable, and narrow regions such as the eyes and blood vessels, making them useful for diagnostics and treatment. A single microrobot requires a specific structure to achieve net displacement in low-Reynolds-number environments. Leveraging this characteristic, researchers use external magnetic fields to drive magnetized microrobots with specialized structures, resulting in a range of propulsion mechanisms. Inspired by natural group migrations, researchers have dynamically assembled multiple microrobots to form various swarms. Swarms have become a significant area in robotics [1]. A coordinated swarm can perform tasks beyond the capability of a single microrobot and significantly enhance task efficiency [2,3].

In low-Reynolds-number microfluidic environments, viscous forces are the primary resistance. Hence, microrobots must break the reciprocity of motion to achieve net displacement [4]. The structural design of a microrobot determines whether it can produce net displacement and what types of propulsion mechanism it can perform. Drawing from scallop theory and inspired by microbial motion, researchers have designed magnetic microrobots with structures such as spiral, spherical, flagellum-like, and multi-joint forms to achieve effective movement [5,6,7,8]. In addition to effective structural design, materials and manufacturing methods are also crucial elements in the study and application of magnetically driven microrobots. Generally, microrobot materials are classified into biological and synthetic categories. Biological materials include various bacteria, cells, and algae, with *E. coli*, red blood cells, and Chlorella being typical examples [9,10]. Biological microrobots are characterized by their natural origin, ease of manufacture, and retention of biological features. Synthetic materials, including various gels like photoresists and hydrogels [11], are noted for their high biocompatibility, degradability, and low cost. Techniques such as direct laser writing (DLW), glancing angle deposition (GLAD), and template-assisted electrochemical deposition (TAED) can be used to fabricate the microrobots desired [12,13,14].

Due to the diminutive size of microrobots (typically on the micrometer-to-nanometer scale), powering them through onboard energy sources is currently challenging. Therefore, various external actuation methods have been proposed, including optical, electrocapillary, ultrasonic, electrostatic, and magnetic drives [15,16,17,18,19]. These methods enable microrobots to navigate within narrow and confined fluidic environments. Compared to other drive mechanisms, the forces and torque generated by magnetic fields offer significant advantages, including non-invasiveness, controllable speed, and good biocompatibility of magnetic materials [20,21,22]. These advantages of magnetically drive microrobots for biomedical applications have led to rapid advancements in this field. Recent successes have demonstrated the immense potential of magnetic microrobots in biomedical applications.

Microrobots exhibit unique fluid dynamic characteristics compared to macroscopic robots, leading to distinctive motion behaviors. Researchers have driven microrobots under external magnetic fields to achieve net displacement in low-Reynolds-number environments. The unique structures of microrobots give rise to various interesting propulsion mechanisms, which can be categorized into three main types: corkscrew-like motion, surface rolling, and ciliary stroke motion [23,24,25]. For instance, spiral microrobots move forward in a ciliary stroke motion under a low-frequency rotating magnetic field, while with increasing frequency of the rotating magnetic field, they gradually advance with a corkscrew-like motion. Spherical microrobots can roll along the walls of a pipe under a rotating magnetic field or tumble under a conical magnetic field. Flagellum-like and multi-jointed microrobots move forward with ciliary stroke motion in an oscillating magnetic field. The diverse propulsion mechanisms of single microrobots lay the foundation for the analysis of swarm motion.

Inspired by bird flock migration in nature, researchers have found that swarms can address large and complex tasks. However, as fundamental mechanisms, swarm–environment interactions, and response mechanisms are still under investigation, adjusting swarms to perform complex and multiple tasks in changing environments remains challenging. It is evident that complex collective movements often depend on simple interactions between individual microrobots. Thus, under external magnetic fields, chaotic individual microrobots interact and dynamically assemble into various swarms. Based on the unique propulsion mechanisms of individual microrobots, swarms can exhibit different propulsion mechanisms, such as aggregation, liquid, vortex, and chains [26,27]. Due to their inherent magnetic forces, magnetic microrobots spontaneously aggregate, and under external magnetic fields, these aggregated microrobots can form a liquid swarm, which provides a new way to study single microrobots within the swarm, form chain-like swarms to perform complex maneuvers in narrow paths and channels, or form vortex swarms to carry cargoes and pick up trash in the surrounding environment, offering new possibilities for wastewater purification.

This review describes the different structures and material advantages of magnetic microrobots, as well as the benefits of magnetic field actuation. It also analyzes the diverse propulsion mechanisms of microrobots and swarms, and highlights the broad application prospects of swarms in fields such as wastewater purification and targeted delivery due to these rich and unique propulsion mechanisms.

## 2. Structural Designs of Microrobots

### 2.1. Spiral Microrobot

#### 2.1.1. Biological Spiral Microrobot

Many microrobot designs draw inspiration from the spiral propulsion behavior of microorganisms in nature, which can facilitate penetration into diseased tissue areas through the use of spiral propulsion methods [28,29,30]. Many bacteria have been used as templates for developing microrobots due to their innate propulsion and sensing capabilities [31,32,33,34]. Zhang et al. [35] developed a magnet-driven microrobot swarm with a spiral structure in 2019, termed ABF. They applied magnetic materials to the microrobots by using electron beam evaporation techniques, as shown in Figure 1a. Precise propulsion and steering can be controlled via a uniform rotating magnetic field. To achieve real-time positioning, Xing et al. [36] used a swarm of spiral anaerobic magnetotactic bacteria containing magnetic proteins as carriers loaded with indocyanine green nanoparticles (INPs). Leveraging their self-propulsion capabilities, these microrobots gradually accumulate toward oxygen-poor tumor regions to perform photothermal therapy, and can be located using fluorescence and magnetic dual-mode imaging techniques. In addition to bacteria, spirulina, a naturally occurring single-cell spiral microorganism capable of photosynthesis and possessing natural fluorescence, has been widely used as a template for microrobots [37,38]. By coating a spirulina swarm with magnetic materials through electroplating or electron deposition processes, controlled propulsion of the swarm can be achieved [39,40,41]. To enable precise positioning of a swarm, Xie et al. [42] introduced a polydopamine (PDA) coating that enhanced the photoacoustic (PA) signal and photothermal effect of a microrobot swarm MSP. Additionally, the intrinsic fluorescence quenching and diverse surface reactivity of PDA allow for fluorescence diagnostics using fluorescent probes.

#### 2.1.2. Synthetic Spiral Microrobot

Naturally, spiral microrobots can be influenced by their biological characteristics, prompting the development of many synthetic spiral microrobots to achieve additional functionalities. Typically, interference lithography or laser printing techniques are used to fabricate spiral robots [43,44,45,46]. Pang et al. [47] utilized holographic lithography (IL) to generate 3D structures by exposing photoresist to optical interference patterns produced by multiple non-coplanar laser beams. This technique can form complex spiral structures at low cost without masks or complex lens systems. By incorporating magnetic materials, targeted delivery of microrobots can be achieved. Considering the complexity of the in vivo environment, Liu et al. [48] developed a hydrogel-based soft spiral microrobot, where the hydrogel exhibits unique adhesion in viscous liquids, enabling adaptive deformation based on the liquid state. By coating the surface of a hydrogel-based soft spiral microrobot with Fe_3_O_4_ particles, the microrobot can navigate through narrow and winding microchannels under the influence of a magnetic field, achieving targeted drug delivery in complex environments. To enhance drug-carrying capacity, Gao et al. [49] modified a spiral structure by adding a drug-carrying cavity and changing the cross section from a traditional circular to an elliptical shape. This structure is expected to achieve higher movement speeds and more efficient drug delivery. To further improve biocompatibility of microrobots for use in in vivo environments, Ceylan et al. [50] designed a microrobot based on enzyme-degradable hydrogel composed of GELMA (gelatin methacryloyl) and superparamagnetic iron oxide particles, as shown in Figure 1b. The microrobot can be degraded by matrix metalloproteinase 2 (MMP-2), which can facilitate the release of embedded cargo molecules. Most spiral microrobots exhibit complex propulsion mechanisms and must be advanced under specific conditions, so it is hard to apply them in complex environments.

### 2.2. Spherical Microrobot

#### 2.2.1. Biological Spherical Microrobot

Cells, an indispensable part of nature, are mostly spherical or near-spherical [51]. Mayorga et al. [52] utilized electron beam evaporation technology to asymmetrically deposit thin metallic layers on one side of defatted sunflower pollen grains, creating a pollen-based swarm (positively charged). The swarm can be targeted to tumors after positively charged drug loading, leading to strong binding and tumor destruction. Beyond cells, spherical microalgae are also used as carriers for developing microrobots due to their excellent properties, such as the ability to perform photosynthesis and good biocompatibility [53,54,55], as shown in Figure 1c. The most commonly used microalga is Chlorella, a unicellular spherical microalga that can produce oxygen through photosynthesis [56]. Chlorella can be used for targeted drug delivery to specific areas after being magnetized. Additionally, Chlorella contains substantial chlorophyll, which can enhance tissue oxygenation and improve hypoxic conditions under specific wavelength light, which is a useful feature for photodynamic therapy [57], as shown in Figure 1d. To mitigate immune responses, microrobots are often functionalized. For example, Qiao et al. [58] covered the surface of Chlorella with a red blood cell membrane. This approach successfully reduced macrophage uptake of the microrobots. Also, Shao et al. [59] applied neutrophil membranes to develop microrobots, leveraging the inherent chemotactic ability of neutrophils. The microrobots were able to autonomously move along chemotactic gradients for targeted drug delivery. However, most self-propulsion mechanisms of biological spherical microrobots are unstable and influenced by the original cell characteristics, which prevents stable movement along a specific trajectory.

#### 2.2.2. Synthetic Spherical Microrobot

Beyond natural biomaterials, many synthetic spherical materials are also used to manufacture microrobots [60,61]. Many spherical microrobots are designed based on the characteristics of Janus microspheres, Janus microspheres are a class of special multiphase, multicomponent functional composite materials with spatially segregated chemical compositions and functional properties. Janus microspheres exhibit a hand-like structure with a hydrophobic concave side and a hydrophilic convex side [62,63], as shown in Figure 1e. The introduction of magnetic components (e.g., iron, nickel) allows for controlled actuation of Janus microspheres [64,65]. Magnetized Janus particles used in spherical microrobots represent a typical design that can perform two different functions simultaneously: remote manipulation via magnetic fields and precise positioning [66]. Under the influence of a magnetic field, Janus microspheres can form various aggregates, such as dimers and trimers [67]. Multiple microspheres and aggregates can combine to form Janus swarms, which can carry more drugs or cells. To enhance biocompatibility, many biocompatible materials (e.g., photoresists, hydrogels) are used in the fabrication of microrobots. Li et al. [68] employed three-dimensional laser lithography to create porous spherical robots and robots coated with nickel for magnetic actuation and titanium for improving biocompatibility. The photoresist and titanium used in this technology can effectively degrade in vivo, eliminating the need for microrobot retrieval.

### 2.3. The Others

In addition to typical spiral and spherical microrobots, flagella and multi-joint microrobots have also been widely researched [69,70]. Some bacteria utilize their flagella and cilia for moving easily [71]. Leveraging this characteristic, Martel et al. [72] designed a swarm of magnetic-controlled bacteria (MTB) as a natural microrobot swarm. Similarly, sperm cells, which have the unique structure of flagella and the innate ability to spontaneously swim towards the ovary, can achieve rapid and effective self-propulsion, making them excellent candidates for microrobot carriers [73,74,75,76,77], as shown in Figure 1f. Xu et al. [78] proposed a drug delivery microrobot based on sperm cells. The motile sperm cell-loaded anti-cancer drugs (doxorubicin) can successfully deliver drugs to tumors. To achieve precise positioning, Magdanz et al. [79] coated bull sperm cells with magnetic nanoparticles to increase the acoustic impedance of the sperm cells and used ultrasound feedback to position the microrobots. Apart from natural biomaterials, some synthetic microrobots can also replicate ciliary stroke motion. For example, Kim et al. [80] first manufactured an artificial cilium-like magnetic microrobot using 3D laser lithography. This microrobot successfully achieved ciliary stroke motion and controlled movement in low-Reynolds-number fluid environments. Inspired by the propulsion mechanisms of flagella, multi-joint microrobots have been developed. Fabrication methods such as self-assembly, electrodeposition, and template-based printing methods are widely used for creating multi-joint microrobots [81,82,83]. Li et al. [84] used a gold segment as the head, two nickel segments as the body, and a gold segment as the tail fin, connected by three flexible silver hinges. Brunet et al. [85] proposed new locomotion gaits for heterogeneous chained modular robots, as shown in Figure 1g. In addition to the above types, rigid microrobots are also widely used. Wang et al. [86] used magnetic responsive materials such as magnetic metals or magnetized nanoparticles as raw materials for microrobots. These materials generate force and torque under the action of a magnetic field, allowing the microrobots to move, rotate, and crawl in a 2D plane. Additionally, microrobots come in various shapes, such as star-shaped and tubular [87,88,89], different shapes can be chosen based on specific needs and environments, and microrobot swarms of varying shapes offer a range of potential applications.

In summary, spherical microrobots are more suitable for rolling on the walls of blood vessels, and spiral microrobots are suitable for propulsion in low-Reynolds-number environments and can easily drill through narrow areas. Flagellum-like microrobots act like the motion of flagella in nature, making them suitable for drug release or cell manipulation in body fluids.

## 3. Actuation

### 3.1. Gradient Magnetic Fields

When an external magnetic field is applied to a magnetic microrobot, it generates corresponding magnetic forces, which can be expressed as [90]:(1)$$F_{m}=V_{m}\left(M\cdot \nabla \right)\vec{B},$$
where $V_{m}$ is the volume of the microrobot, B is the magnetic flux density of the magnetic field, and M is the magnetization strength of the object. From Equation (1), it can be inferred that the applied magnetic force is proportional to the magnetic field gradient, which means that the magnetic force depends on the gradient of the magnetic field, allowing the microrobot to be guided from areas of low to high magnetic force.

The gradient magnetic field can be formed by a single permanent magnet or multiple permanent magnets superimposed [91], as shown in Figure 2a. Initially, researchers used single-permanent-magnet systems to actuate microrobots, where the size and orientation of the permanent magnet were core factors of the actuation system. By adjusting the position or orientation of the permanent magnet, the microrobot could be moved in different directions under the influence of the magnetic gradient, typically achieving one-degree-of-freedom (DOF) control [92]. To achieve more complex motions, multiple permanent magnet systems with specific arrangement distributions can be used to increase DOF. For instance, a four-magnet system composed of two pairs of symmetrical permanent magnets along the x–z and y–z planes can successfully achieve three-DOF control [93]. However, the magnetic field generated by permanent magnets cannot be changed in real time and lacks precise control over the magnitude of the magnetic field. To address this issue, gradient magnetic fields generated by electromagnetic coils are widely used [94,95]. Maxwell coils are often employed to produce a uniform magnetic field gradient, described by the gradient Gm as follows [96]:(2)$$G_{m}=\frac{16}{3}{\left(\frac{3}{7}\right)}^{5/2}\frac{\mu_{0}NI}{a^{2}},$$
where $\mu_{0}$ is the magnetic permeability, N is the number of turns in the coil, I is the current, and a is the radius. According to Equation (2), it can be seen that the strength of the gradient is influenced by the number of coil turns and the current passing through the coil. Therefore, researchers have enhanced the magnetic gradient by increasing the number of turns in the coil [97] or designing magnetic cores [98], achieving gradients up to 20 T/m. By precisely controlling the magnetic gradient, more accurate magnetic force control can be achieved. However, in practical applications, driving is often accomplished by paired coils [99,100], where a pair of Maxwell coils can only achieve one-DOF manipulation. To improve a system’s DOF, distributed electromagnetic systems are proposed. Typically employing six or eight orthogonal coils evenly distributed around the workspace, each electromagnetic coil pointing towards the center can generate a strong magnetic gradient [97,101,102,103], as shown in Figure 2b,c. The most famous one is a system called OctoMag, which has eight electromagnets divided into two sets: the upper set has an angle of 45° with respect to the axis of symmetry, while the lower set is 90°. Also, the upper set has a 45° rotation relative to the lower set along the axis of symmetry [104]. The wireless operation of three-DOF locomotion and two-DOF orientation is achieved through this distribution.

Compared with the other magnetic fields, the gradient magnetic field has fewer restrictions on the mechanical structure of the microrobot, such as spiral [105], spherical [106], or other shapes, which is the advantage of the gradient magnetic field.

### 3.2. Rotating Magnetic Fields

In a rotating magnetic field, the microrobot is regarded as a magnetic dipole model relative to the external magnetic field, using the magnetic moment m and the magnetic flux density B to describe its potential energy U, as shown in Equation (3). The magnetic moment T of the magnetic dipole can be expressed by Equation (4). When m and B are vertical, T is at its maximum, and B is zero when m is parallel to B. Due to the alignment trend of the internal magnetization and the magnetic field, the microrobot rotates under the influence of the external rotating magnetic field [107,108]:(3)U=−m·B,(4)T=−m×B,

A rotating magnetic field can also be generated by a single permanent magnet, which is usually mounted on the mechanical arms [109,110], as shown in Figure 2d. By rotating the mechanical arm, the permanent magnet can be rotated along the axis, thus generating a rotating magnetic field to provide additional DOF. Researchers manipulated the mechanical arm to control the permanent magnet to drive a microrobot, successfully achieving two locomotion degrees of freedom and three orientation degrees of freedom. However, spontaneous magnetic gradients produce unwanted lateral magnetic forces, often affecting the drive of the rotating magnetic field. Therefore, multi-magnet systems that can reduce the influence of the magnetic gradient are proposed [111,112,113], as shown in Figure 2e. Due to the symmetric distribution of permanent magnets, the magnetic field gradients will cancel each other, among which the most typical is the system called Niobe, which is composed of two relatively distributed permanent magnets. Each permanent magnet can rotate, and thus the influence of the magnetic gradient is significantly reduced [114]. However, the size of the rotating magnetic field generated by the permanent magnet cannot be precisely regulated, and the accuracy problem remains. Therefore, electrified Helmholtz coils are often used to generate a uniform magnetic field, providing orientation degrees of freedom. By changing the current in the coil, the size and direction of the rotating magnetic field can be controlled and the accuracy of the control can be improved. The most common is a magnetic field generating system composed of three sets of orthogonally arranged Helmholtz coils. The resulting magnetic field can be expressed as [115]:(5)$$B_{⊥n}\left(t\right)=B_{0}COS\left(2\pi ft\right)\tilde{u}+B_{0}COS\left(2\pi ft\right)\tilde{v},$$
where $B_{0}$ represents the magnetic flux density at the center of the Helmholtz coil, f represents the rotation frequency, and $\left(\tilde{u},\tilde{v}\right)$ represents the plane vector orthogonal to the axis n. The system above is commonly used to generate a rotating magnetic field.

In a rotating magnetic field, microrobots with special structures exhibit unique propulsion mechanisms. Due to the effective movement and translation of microrobots in low-Reynolds-number fluid environments under the drive of a rotating magnetic field, they are well suited for internal applications, and this is one of the advantages of using a rotating magnetic field [116].

### 3.3. Oscillating Magnetic Fields

An oscillating magnetic field refers to an electromagnetic field that rapidly alternates in polarity or direction over time. In the field of microrobots, this phenomenon has significant application value. An oscillating magnetic field is typically used in conjunction with other uniform magnetic fields. The uniform field aligns the microrobot with the magnetic field without pulling the microrobot along the field gradient, thus ensuring that the microrobot’s movement is solely due to the oscillating magnetic field [117,118,119]. For example, He et al. [120] generated an oscillating magnetic field using an orthogonal array of electromagnetic coils, as shown in Figure 2f. This system is designed to drive flagellated microrobots, with two electromagnets producing the oscillating magnetic field, causing the microrobots to move in opposite directions, while the other two electromagnets provide a uniform magnetic field to establish direction. However, the oscillating magnetic field produced by this system is symmetrical. Qiu et al. [121] proposed that if the oscillating magnetic field is symmetrical, such as a sine wave, the net forward displacement of flagellum-like microrobots will be zero, whether in thickening or thinning fluids. Additionally, asymmetric driving in Newtonian fluids also leads to zero net displacement. Therefore, to enable such microrobots to achieve forward displacement, asymmetric oscillating magnetic field generation systems have been proposed. Yang et al. [122] successfully developed a self-oscillating generator by utilizing asymmetrically induced ordered physical mechanisms, as shown in Figure 2g. The system’s arrangement symmetry was intentionally broken to achieve continuous ciliary stroke motion, and the generator can provide power to onboard microrobot components through interactions of simple particles. Oscillating magnetic fields enable microrobots with specialized structures to perform various types of propulsion, such as ciliary stroke propulsion and surface walking propulsion (discussed in detail later), holding considerable potential in biomedical and biosensing research [84,123].

In summary, a gradient magnetic field guides microrobots of any shape along the magnetic field gradient; a rotating magnetic field enables corkscrew-like motion, suitable for propulsion in low-Reynolds-number environments, but requiring complex design; and an oscillating magnetic field drives microrobots with specific structures, but with lower control of accuracy and stability.

## 4. Propulsion Mechanisms of Single Microrobot

### 4.1. Corkscrew-like Motion

Under the influence of a rotating magnetic field, spiral microrobots can perform corkscrew-like motion. As described in Equations (3) and (4), the dipole moment of the microrobot continually aligns with the rotating magnetic field, and the magnetic torque causes the microrobot to rotate, generating a forward-propelling force [35,124,125]. In low-Reynolds-number environments, viscous forces dominate and reciprocating motion cannot propel the microrobot; however, spiral microrobots can avoid reciprocating motion and achieve forward displacement under the influence of the magnetic field [126]. The movement of the microrobot is influenced by various factors, such as the pitch angle between the microrobot and the spiral axis and the frequency of the magnetic field. When a spiral microrobot is placed in a rotating magnetic field, it will stably rotate around the spiral axis. The robot’s direction of movement can be altered by controlling the direction of the applied magnetic field (clockwise or counterclockwise), allowing it to advance or retreat in a spiral manner [49,127,128], as shown in Figure 3a,b. The propulsion efficiency of the spiral microrobot is highest when the microrobot is at a 45° angle to the spiral axis [129]. However, low-frequency rotating magnetic fields (typically below a few hertz) can cause wobbling motion when the spiral axis fails to align with the local magnetic field direction [130]. With increasing frequency, the spiral axis gradually aligns with the magnetic field direction, and the spiral microrobot returns to corkscrew-like motion. This mode of movement allows the robot to navigate easily in high-viscosity fluids, making it particularly suitable for in vivo applications.

### 4.2. Spherical Microrobots

Powerful fluid flow presents a significant obstacle to controlling navigation in vivo, and spiral propulsion is more suitable for vessels with lower flow rates. Therefore, a rotating magnetic field is employed to drive spherical microrobots to roll along the container surface wall, effectively mitigating the issue of excessive liquid flow by utilizing the principle of lower flow rates at the container wall [131,132]. The internal magnetic moment of the spherical microrobot continually changes and reorients due to the external magnetic field, generating a torque that initiates rotation. As the microrobot is spherical, it continues to roll [133]. Through this propulsion mechanism, Yang et al. [134] successfully enabled spherical microrobots to roll on periodically raised surfaces. Jeon et al. [135] proposed a novel self-positioning and rolling magnetic microrobot (SPRMM) actuated by a magnetic navigation system, as shown in Figure 3c. This “surface rolling” propulsion mechanism allows spherical microrobots to traverse along vessel walls in complex internal environments, making them more suitable for moving in vivo applications. Under the influence of the magnetic field, the rolling spherical robots interact and connect. Li et al. [67] invented a microrobot named the Janus micro-dimer surface walker composed of two Janus spheres, which is capable of periodic rolling under an external oscillating magnetic field. Unlike independent rolling, the dimer rolls in a coordinated and hinged manner. This kind of microrobot is usually defined as dimer [136], as shown in Figure 3d. Spherical microrobots can exhibit various propulsion mechanisms under different external magnetic fields, making them well suited for dynamic environments.

### 4.3. The Others

Similar to spiral microrobots and inspired by the swimming propulsion mechanisms of fish or bacterial flagella, researchers have proposed ciliary stroke motion as an effective propulsion mechanism for microrobots [137,138]. When these flagellum-like microrobots are subjected to an oscillating magnetic field, their heads vibrate, causing the microrobots to propel forward with a ciliary stroke motion akin to sperm [139,140]. For example, Magdanz et al. [141] revealed distinct flow fields, propulsive thrust, and frequency responses during flagellar propulsion, as shown in Figure 3e. Besides flagellum structures, multi-joint configurations can also achieve ciliary stroke motion under magnetic fields. Vyskocil et al. [142] proposed Au/Ag/Ni microrobotic scalpels controlled by a transversal rotating magnetic field, which can enter the cytoplasm of cancer cells and are also able to remove a piece of the cytosol while leaving the cytoplasmic membrane intact in a microsurgery-like manner, as shown in Figure 3f. Similarly, Wu et al. [143] developed a multi-joint serpentine microrobot. They demonstrated the smooth forward movement of the developed microrobot. Based on theoretical analysis and experiments, at least three segments were needed, and microrobots with four to five segments were required for optimal performance. These microrobots with ciliary stroke motion hold extensive potential for navigation and targeted tasks in complex environments, as detailed in Table 1.

## 5. Propulsion Mechanisms of Swarm

### 5.1. From Single Microrobots to Swarm

The propulsion mechanisms of different microrobots are diverse and varied. If a sequence of moving microrobots can be programmed, a group of microrobots can dynamically change formations according to tasks, similar to the collective behavior of small animals like ants [144]. To achieve dynamic assembly of multiple single microrobots into a target swarm, it is essential to prevent the magnetic microrobots from attracting each other and forming irregular swarms when they come close, and thus their motion must be constrained to ensure that single microrobots within a swarm always repel each other [145]. Additionally, the balance of positions of these microrobots must be programmable and reconfigurable. Typically, different external magnetic fields are used to allow microrobots to assemble as desired. By designing the external magnetic field distribution, their balance positions can be precisely programmed to achieve dynamic assembly and disassembly [146,147]. Under the influence of external magnetic fields, single microrobots exhibit various propulsion mechanisms (e.g., corkscrew-like motion, rolling, ciliary stroke motion), and it is obvious that large-scale collective behavior of the swarms largely depends on these simple interactions. Dynamic changes in single microrobots can trigger the emergence of various forms of self-organized swarm, such as aggregation, liquid, vortex, and chain, as detailed in Table 2. Moreover, animal swarms can reconfigure their formations based on tasks, such as foraging and transportation, allowing different functions within the same group. Inspired by behaviors of animal swarms, freely dynamically assembled and disassembled swarms have been developed for various functions [148,149], such as water purification and targeted delivery.

### 5.2. Liquids

In the absence of an external magnetic field, magnetized microrobots will spontaneously assemble together due to their inherent magnetic dipole forces, forming a random swarm [150,151]. To analyze the forces and mechanisms of single microrobot within the swarm, they often need to be separated into a uniformly distributed propulsion mechanism, known as the liquid [152,153]. Fan et al. [154] proved that a swarm composed of ferrofluid droplets are inherently deformable and can be controlled individually or in aggregate (liquid or chain), as shown in Figure 4a. Similarly, under a low-frequency rotating magnetic field, each spiral microrobot in a swarm displaces in the direction of its spiral propulsion. Wang et al. [37] demonstrated that their swarm achieved an efficient propulsion performance with a maximum speed of 526.2 μm/s under a rotating magnetic field. Additionally, they could disintegrate into individual particles under near-infrared laser illumination. Huang et al. [155] enabled microrobot structure transition from an initial folded state to a form with complementary spiral angles, a reversible mechanism transition that can facilitate effective drug-targeted delivery in complex environments (such as narrow pores), as shown in Figure 4b. The liquid of the swarm makes it possible to address single microrobots within the swarm [135].

### 5.3. Vortices

Under the influence of an external rotating magnetic field, a swarm will form a small vortex at the rotating core [156], as shown in Figure 4c. This rotation induces dynamic interactions with the surrounding fluid. Concurrently, magnetic dipole forces between microrobots cause them to approach each other, while short-range repulsive forces prevent them from getting too close, resulting in spatial arrangement and interaction. As more microrobots accumulate, a stable vortex flow field is formed. With increasing microrobot density, phase separation occurs within the system and the vortex gradually enlarges [157,158]. The size of the vortex formed by the swarm is influenced by several factors [159,160]. Higher microrobot density tends to facilitate the formation of larger vortices. Additionally, the rotational frequency and intensity of the external magnetic field directly impact microrobot motion and interactions, which in turn affect the generation and stability of vortices. By adjusting parameters of the external magnetic field, such as rotational direction and frequency, the direction and morphology of the vortex can be regulated, demonstrating good manipulability and controllability. Over time, two adjacent vortices with the same polarity can merge into a larger vortex [27]. Wang et al. [161] utilized vortex swarms composed of magnetic nanoparticles for large-scale targeted drug delivery, as shown in Figure 4d. These swarms can withstand high-flow impacts, move upstream along container walls, and settle at target locations. When swarms encounter tumor peripheries lacking continuous vessel walls, their rheological properties actively cause them to adhere to the edges of endothelial gaps, using their deformability to traverse narrow intercellular spaces. The vortex propulsion mechanism enables a swarm to carry substantial amounts of drug cargoes, offering significant prospects for targeted delivery applications.

### 5.4. Chains

In the presence of a low-frequency rotating magnetic field, spherical microrobots undergo a surface-rolling propulsion mechanism. When two adjacent rolling spherical microrobots interact, fluid dynamic interactions and magnetic dipole forces create a dynamic equilibrium, resulting in the formation of a stable pair of rotating microrobots [162], known as a dimer (see Section 3.2). The rotating magnetic field can capture new microrobots to join a chain, making it longer and more complex [163], as shown in Figure 4e. Additionally, Chen et al. [8,164,165] experimentally analyzed and verified the formation, motion, and integrity of chain-like swarms in fluid flow, and the velocity of the chain-like swarms increased with the number of microrobots. Wang et al. [166] proposed that if the number of microrobots reaches a certain threshold (approximately six microrobots), the rate of velocity increases, gradually diminishes, then stabilizes. Due to the elongated shape, chain-like swarms are particularly well suited for traversing narrow, long blood vessels, offering potential for targeted drug delivery and removal of blockages in constricted channels.

**Table 2 micromachines-16-00181-t002:** The propulsion mechanisms of swarm.

Propulsion Mechanism	Type of Magnetic Field	Structure of Microrobot	Reference
Liquid	Rotating	Spiral/flagellum-like/multi-joint	[37]
	Oscillating	Spherical	[153]
Vortex	Rotating	Spherical/spiral/flagellum-like/multi-joint	[161]
Chain	Rotating	Spherical	[166]

## 6. Applications of Magnetic Swarms

### 6.1. Water Purification

Utilizing magnetic microrobots or swarms in water purification is advancing rapidly. When subjected to an external magnetic field, microrobots can interact with target pollutants, thereby enhancing and accelerating the water purification process [167]. Research has demonstrated that miniature vortices generated by rotating objects are capable of capturing and transporting microparticles, suggesting that fluid vortices can facilitate particle transport without mechanical contact [168,169], as shown in Figure 5a. Under a rotating magnetic field at a specific frequency, vortex swarms are formed, which are highly effective in capturing and sequestering pollutants in water. Sun et al. [170] used a “hedgehog” spherical swarm based on sunflower pollen grains (SPGs) to capture and store oil pollutants, and it could also adsorb oil droplets by capillary action. The fluid dynamics generated by collective behavior can remove microplastics from wastewater without mechanical contact, as shown in Figure 5b. Also, under the control of an external magnetic field, the heads of spiral microrobots will form a series of small vortices, which is highly beneficial for the development of water purification work. Chan et al. [171] designed a spirulina-based magnetically controlled hollow swarm with high specific surface area (SSA), meaning more active sites on the surface for reaction or adsorption. Under the action of a rotating magnetic field, heavy-metal ions are collected using gradient concentrations, and at the same time, because the swarm contains magnetic nanoparticles, photothermal effects can be used to kill bacteria, as shown in Figure 5c. The vortex swarm actively seeks and removes pollutants in water through remote magnetic field control, and has the characteristics of recyclability, biological compatibility, and stimulus response, which brings revolutionary improvements to traditional water purification methods and has broad application prospects.

### 6.2. Targeted Delivery

Wireless propulsion and controlled motion make swarms excellent candidates for targeted delivery and therapy. Swarms can not only deliver drugs or cells to the desired location but also improve the efficacy of treatment [172], as shown in Figure 5d. During the process of targeted drug delivery, drug loss is inevitable. Spherical swarms can carry a large number of drugs, and large spherical swarms can be separated into different numbers of small swarms, while small spherical swarms can also be merged into large spherical swarms. Based on this feature, spherical swarms are very suitable for delivering drugs in the complex environment within the body and ensuring a sufficient dosage of drugs is delivered [173,174]. By programming the magnetic field parameters, Wang et al. [175] showed that a vortex swarm is variable and reconfigurable, able to easily overcome the complex surface environment composed of biological cells and control the movement of the surface in any directions. What is more, Martel et al. [176] successfully achieved targeted drug delivery using magneto-aerotactic bacteria swarm guided by magnetic fields to move to the hypoxic region of a tumor and enrich the hypoxic region of the tumor. However, in the large vascular system, diminutive vortex swarms will move together and become a large swarm, with high speed, driving force, and rate of movement, and thus it is difficult to control such a swarm in confined environments, especially in the blood circulation system [177]. Because of the narrow and long characteristics of the chain-like swarm, it is especially suitable for passing through narrow and long blood vessels. Using the propulsion mechanisms of chain-like swarms [178]. Lu et al. [179] proposed a multi-level magnetic microrobot that can transport within a hierarchical vascularized organ-on-a-chip, as shown in Figure 5e. In addition to improving drug loading [180], Villa et al. [181] used weak magnetic fields to drive chain-like swarms to capture and transport cancer cells, as shown in Figure 5f. Xu et al. [182] proposed the Hercules swarm to achieve rapid delivery (100 mm s^−1^), efficient cargo transport (carrying twice its own weight), and effective catheter clearance (1 mm min^−1^), as shown in Figure 5g. Swarms demonstrate multiple propulsion mechanisms and morphological change capabilities in confined spaces due to their small dimensions, making them promising candidates for microrobot manipulation and targeted delivery in a non-invasive manner.

## 7. Summary and Prospects

In recent decades, the study of microrobots has achieved significant advancements. Magnetically actuated microrobots offer advantages such as fast response, remote and non-contact control, harmless human–machine interaction, and cost-effectiveness. Under external magnetic driving strategies, these microrobots can achieve precise manipulation and navigation in extreme biomedical environments. Inspired by the propulsion mechanisms and collective behaviors of microscopic organisms in nature, researchers have developed various complex and programmable swarms. Magnetic swarms exhibit multiple propulsion mechanisms and possess remarkable capabilities in areas such as wastewater purification, targeted delivery, and precise operations.

This paper reviews various propulsion mechanisms of magnetic microrobots and swarms, as well as the characteristics of different magnetic fields commonly used. The structural design of microrobots is crucial in determining their movement under external drives. Common structures include spiral, spherical, flagella, and multi-joint designs. The design templates for microrobots draw inspiration from natural microorganisms, such as spiral bacteria, flagellated sperm, and various-shaped algae, which possess inherent propulsion and sensing capabilities. To achieve additional functionalities, researchers have used a variety of synthetic materials like hydrogels, which offer good degradability. Manufacturing microrobots from synthetic materials are complex and require advanced techniques such as interference lithography and laser printing to achieve intricate structural designs. Coating microrobots with magnetic nanoparticles allows for controlled movement under external magnetic fields. Gradient magnetic fields can drive microrobots of various shapes, while rotating magnetic fields can induce helical microrobots to perform bottle opening-like motions and spherical microrobots to roll. Oscillating magnetic fields can drive flagellar and multi-joint microrobots to perform fan-like oscillations. Under the influence of external magnetic fields, these robots exhibit diverse propulsion mechanisms to accomplish various tasks, thereby advancing the clinical applications of micro/nanorobots.

Despite substantial progress in research on magnetically driven microrobots and swarms, there are still several challenges. Most research has focused on spherical microrobot swarms, with less attention given to swarms of other shapes. While motion control technology for individual magnetic microrobots is well developed, control of magnetic swarms is still an emerging field, lacking a systematic theoretical framework. Further simulation of natural group migration, reconfigurability, adaptability, and 3D collective behaviors is needed, along with expanding the operational dimensions of microrobot swarms and enhancing stability in complex environments. Addressing these issues requires a better understanding of cluster movement principles, mechanisms, and designs. From an application perspective, particularly in biomedicine, it is crucial to consider the effects of dynamic internal environments (such as flowing fluids, pulsating boundaries, and irregular surface morphologies) and the safety of swarms. Future research should address these challenges comprehensively, aiming to develop swarms capable of precise task execution through collaborative operations and autonomous decision-making via artificial intelligence, presenting promising research prospects.

## Figures and Tables

**Figure 1 micromachines-16-00181-f001:**
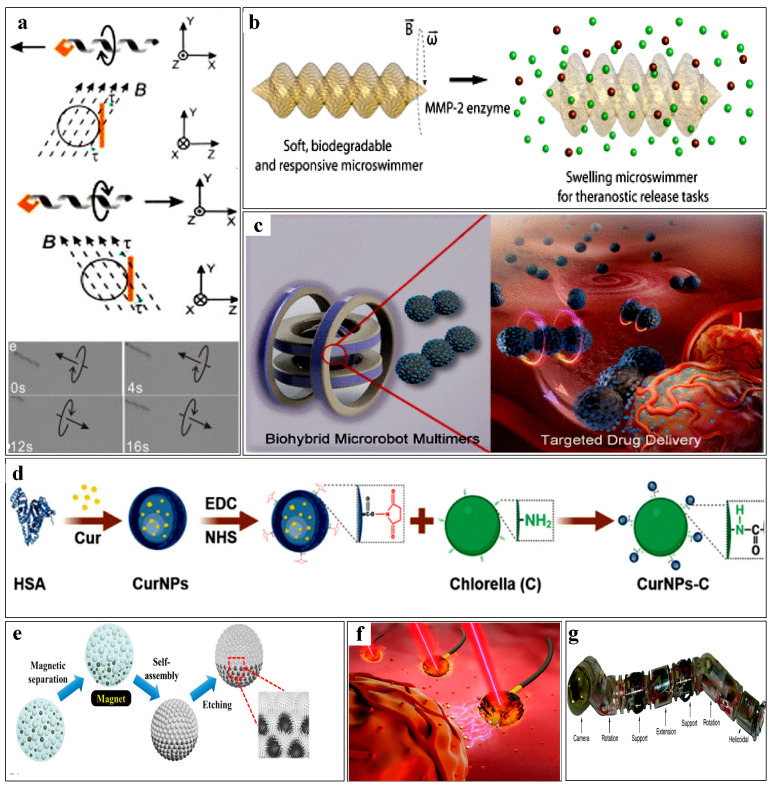
Structural design of microrobots. (**a**) Image of a magnetic spiral microrobot. (**b**) Schematic of a hydrogel-based artificial double-spiral microrobot responding to pathological concentrations of MMP-2 through swelling and releasing drugs. (**c**) Schematic of a magnetic biohybrid microrobot based on Chlorella for targeted drug delivery. (**d**) Overlaying CurNPs on the surface of Chlorella. (**e**) Microrobot based on Janus microspheres. (**f**) Flexible flagellated microrobot based on sperm cell for targeted chemical photothermal therapy. (**g**) Image of an artificial microrobot with a multi-segment structure.

**Figure 2 micromachines-16-00181-f002:**
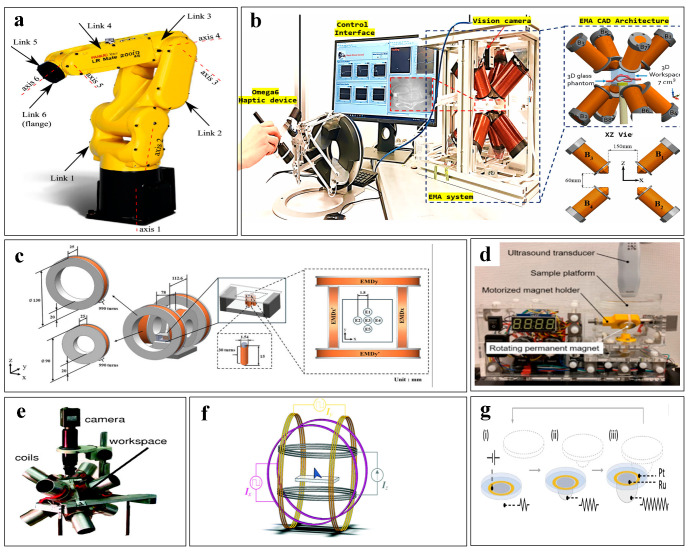
Systems of magnetic fields for actuating microrobots. (**a**) A system based on magnets producing a gradient magnetic field. (**b**) A distributed electromagnetic coil system with eight coils. (**c**) A distributed electromagnetic coil system capable of generating five-degrees-of-freedom control. (**d**) A magnet mounted on a robotic arm to increase the system’s degrees of freedom. (**e**) Multiple magnets placed on a robotic arm to generate a rotating magnetic field. (**f**) Electromagnetic coils symmetrically arranged to produce a symmetrical oscillating magnetic field and a uniform magnetic field. (**g**) A circuit for generating an asymmetric oscillating magnetic field.

**Figure 3 micromachines-16-00181-f003:**
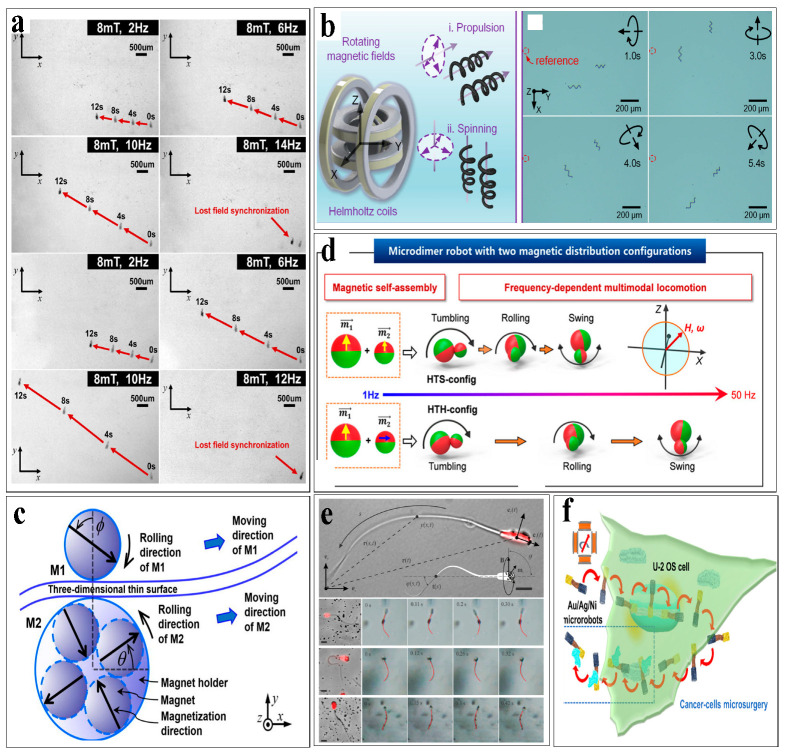
Propulsion mechanisms of single microrobots. (**a**) A spiral microrobot performing a corkscrew-like motion under a rotating magnetic field. (**b**) A spiral microrobot moves under a magnetic field. (**c**) Spherical microrobot rolling on an uneven surface under an external rotating magnetic field. (**d**) Spherical dimers moving forward like fish under an oscillating magnetic field. (**e**) Flagella-like microrobot moving forward under an external oscillating magnetic field. (**f**) A multi-joint microrobot swinging like a fish under external magnetic field.

**Figure 4 micromachines-16-00181-f004:**
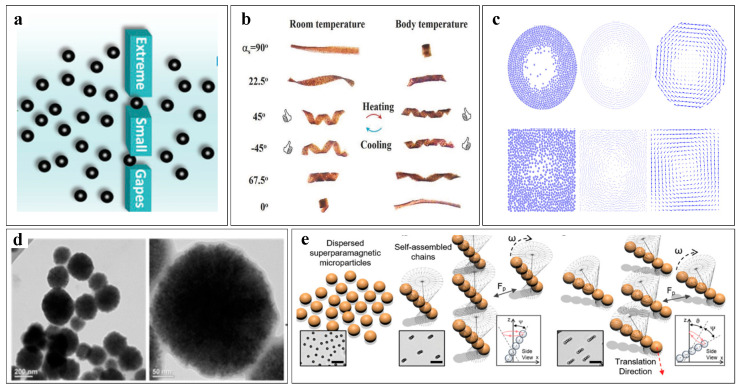
Multiple propulsion mechanisms of microrobot swarms. (**a**) Under the influence of an external magnetic field, spherical particles are distributed uniformly. (**b**) Under the effect of an external alternating magnetic field, single spiral microrobots within the swarm undergo a mode transition. (**c**) Magnetic particles gradually form vortex swarms under the control of an external magnetic field. (**d**) Scanning electron microscopy (SEM) image of a vortex swarm composed of magnetic particles. (**e**) Magnetic Janus microspheres form chain-like swarms under the influence of an external low-frequency rotating magnetic field.

**Figure 5 micromachines-16-00181-f005:**
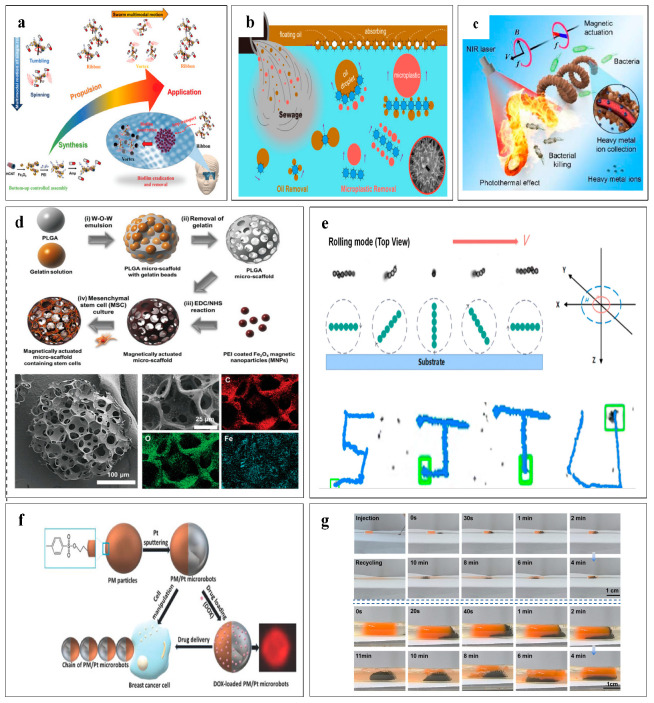
Applications of microrobot swarms. (**a**) Vortex swarm and antibacterial diagram. (**b**) Vortex swarm based on sunflower powder particles (SPG) for oil and microplastic pollution removal. (**c**) Spirulina-based swarm for photothermal antimicrobial therapy and heavy-metal ion collection. (**d**) Multi-void spherically based magnetized swarm for targeted delivery of mesenchymal stem cells. (**e**) Magnetic chain swarm moving along a special path. (**f**) Schematic diagram of a magnetized chain swarm capturing and transporting cells under the action of an external rotating magnetic field. (**g**) A chain swarm clears debris from the pipeline under the drive of an external magnetic field.

**Table 1 micromachines-16-00181-t001:** Propulsion mechanisms of single microrobots.

Propulsion Mechanism	Type of Magnetic Field	Structure of Microrobot	Reference
Corkscrew-like motion	Rotating	Spiral	[127]
Ciliary stroke	Rotating	Spiral	[130]
	Oscillating	Spherical dimer/flagellum-like/multi-joint	[139]
Surface rolling	Rotating	Spherical	[133]

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
