# Peer review of "Propulsion Mechanisms in Magnetic Microrobotics: From Single Microrobots to Swarms"

_micromachines, 2025, doi:10.3390/mi16020181_

Round 1
Reviewer 1 Report
Comments and Suggestions for Authors
This review paper provides a comprehensive and insightful exploration of the relationship between single microrobots and their swarms, focusing on structural design, magnetic field control strategies, propulsion mechanisms, and practical applications in complex environments.
It is exceptionally well-organized, with thorough and forward-thinking discussions on key topics such as structural design, propulsion mechanisms, and real-world applications. Overall, this polished and valuable contribution to the field requires no revisions.
Author Response
Thanks for your encouragement comments!
Reviewer 2 Report
Comments and Suggestions for Authors
Comments to the author
======================
Review of "Propulsion mechanisms in magnetic microrobotics: From single microrobot to swarm"
This paper presents the relationship between single microrobot and swarm by introducing the structural design of microrobots and magnetic field generation systems, analyzing the propulsion mechanisms of individual microrobot under the external magnetic field, and discussing the potential applications and challenges.
Here are some specific comments:
Questions & Suggestions:
1. In lines 62-63, as author describes the “external actuation methods”, the electrocapillary actuation is suggested to be added into this category. However, Chemical actuation is widely regarded as self-propelled method instead of external actuation method.
2. In the parts of 1.3 and 3.3, In addition to the mentioned structures of microrobots, 2D rigid microrobots were also reported, and they can be actuated by processing magnetic field. It’s suggested to enrich the content.
3. In line 211, words are missing in the sentence, and please check the grammar of the sentence in line 108.
4. Please check and modify the image resolution (including text notes), especially in Figures 1 and 5.
Author Response
Comments 1: [In lines 62-63, as author describes the “external actuation methods”, the electrocapillary actuation is suggested to be added into this category. However, Chemical actuation is widely regarded as self-propelled method instead of external actuation method.] Response 1:Thank you for pointing this out. l agree with this comment. Therefore, l have removed the imprecise term "chemical drive" and added “electrocapillary drive” along with the relevant reference [19], as detailed in lines 60 and 61.
Comments 2: [ In the parts of 1.3 and 3.3, In addition to the mentioned structures of microrobots, 2D rigid microrobots were also reported, and they can be actuated by processing magnetic field. It’s suggested to enrich the content.]
Response 2:Thank you for pointing this out. I agree with this comment. Therefore, I have added the following relevant content: [In addition to the above types, rigid microrobots are also widely used,. Wang et al.[86]used magnetic responsive materials such as magnetic metals or magnetized nanoparticles as raw materials for microrobots. These materials will generate force and torque under the action of a magnetic field, allowing the microrobots to move, rotate, and crawl in a 2D plane. ], as shown in line 217-220.
Comments 3: [In line 211, words are missing in the sentence, and please check the grammar of the sentence in line 108.]
Response 3:Thank you for pointing this out. I have already added the correct word "flagella", as detailed in line 210. While the sentence in line 108 has been revised, the correct sentence is as follows:[Many microrobots draw inspiration from the spiral propulsion behavior of microorganisms in nature, which can facilitate penetration into diseased tissue areas through the use of spiral propulsion methods.], as shown in line 105.
Comments 4: [Please check and modify the image resolution (including text notes), especially in Figures 1 and 5.]
Response 4: Thank you for pointing this out. I have replaced all the images in the manuscript with high-resolution ones. Please check the new version of the manuscript for details.
Reviewer 3 Report
Comments and Suggestions for Authors
In this review, the authors conduct a comprehensive analysis of the classification, fabrication methodologies, and characteristics of various magnetic fields relevant to microrobotics. They delve into the propulsion mechanisms employed by individual microrobots as well as swarms, elucidating the interrelationship between the two. While this review holds significant academic merit, certain aspects warrant further refinement. Below are my detailed comments and suggestions:
a)The review presents a systematic overview of the fabrication techniques for microrobots, yet it lacks an in-depth discussion on the comparative advantages and limitations of the various approaches.
b)Although the review categorizes microrobots by their structural configurations, it would be pertinent to investigate which geometries (e.g., spiral, spherical) are most effective for specific application contexts.
c)The discussion on different magnetic field generation methods is valuable, but an analysis detailing the pros and cons of each magnetic field type is necessary to provide a more rounded understanding.
d) The section addressing the propulsion mechanisms of individual microrobots and swarms could benefit from a more nuanced categorization of these mechanisms.
e)Within the discussion of propulsion mechanisms, it would be advantageous to explore criteria for selecting the optimal propulsion strategy tailored to various application scenarios.
f)The treatment of swarm propulsion mechanisms is somewhat cursory; a more thorough examination of coordination strategies among swarms and an analysis of collaborative behaviors would enhance this section significantly.
Comments on the Quality of English Language
Improvements are still needed.
Author Response
Comments 1:[The review classifies microrobots based on their structure, but it would be beneficial to further explore which structures (e.g., spiral, spherical) are more suitable for different application scenarios.]
Response 1:Thank you for pointing this out. I agree with this comment. Therefore, I have added the following relevant content: [In summary, spherical microrobots are more suitable at rolling on the walls of blood vessels, and spiral microrobots are suitable for propulsion in low Reynolds number environments and can easily drill through narrow areas. While flagella-like microrobots act like the motion of flagella in nature, making them suitable for drug release or cell manipulation in body fluids.], as shown in line 224-228.
Comments 2:[The review discusses the generation of different magnetic fields, but the advantages and disadvantages of various types of magnetic fields is needed.]
Response 2:Thank you for pointing this out. I agree with this comment. Therefore, I added the following related content: [In summary, the gradient magnetic field guides microrobots of any shape along the magnetic field gradient; the rotating magnetic field enables corkscrew-like motion, suitable for propulsion in low Reynolds number environments but requiring complex design; the oscillating magnetic field drives microrobots with specific structures, but with lower control accuracy and stability.], as shown in line 344-348.
Comments 3:[The review mentions the propulsion mechanisms of individual microrobot and swarms, but these propulsion mechanisms could be further categorized.]
Response 3:Thank you for pointing this out. I agree with this comment. Therefore, I have managed the following relevant content: [The propulsion mechanisms of individual microrobots and swarms could be further categorized: the motion modes of individual microrobot can be divided into cork-screw motion, surface rolling motion, and ciliary stroke motion, while the motion modes of swarms can be divided into liquid, vortex, and chain.], as shown in line359, 390, 402, 447, 463 and 484.
Comments 4:[In the propulsion mechanisms section, it would be valuable to examine how to select the most appropriate propulsion mechanisms for different application scenarios.]
Response 4:Thank you for pointing this out. I agree with this comment, these information can be found in line 461, 482 and 495.[The Liquid of the swarm makes it possible to address single microrobot within the swarm; The vortex propulsion mechanism enables the swarm to carry substantial amounts of drug cargoes, offering significant prospects for targeted delivery applications; Due to the elongated shape, chain-like swarms are particularly well-suited for traversing narrow, long blood vessels, offering potential for targeted drug delivery and removal of blockages in con-stricted channels. ]
Comments 5:[The review mentions propulsion mechanisms of swarms, but it provides limited discussion on the strategies for coordination among swarms, a deeper analysis of swarm collaboration could be added.]
Response 5:Thank you for pointing this out. I agree with this comment. Therefore, I have revised the following relevant content: [Typically, different external magnetic fields are used to allow microrobots to assemble as desired, by designing the external magnetic field distribution, their balance positions can be precisely programmed to achieve dynamic assembly and disassembly[146,147]],as shown in line 437.